# The Role of Dental Practitioners in the Management of Oncology Patients: The Head and Neck Radiation Oncology Patient and the Medical Oncology Patient

**DOI:** 10.3390/dj11050136

**Published:** 2023-05-17

**Authors:** Adepitan A. Owosho, Katherine DeColibus, Beverly Hedgepeth, Burton C. Wood, Ritter E. Sansoni, John P. Gleysteen, David L. Schwartz

**Affiliations:** 1Department of Diagnostic Sciences, College of Dentistry, The University of Tennessee Health Sciences Center, Memphis, TN 38163, USA; 2Department of Otolaryngology—Head & Neck Surgery, College of Medicine, The University of Tennessee Health Sciences Center, Memphis, TN 38163, USA; 3Division of Oral Diagnosis, Department of Diagnostic Sciences, College of Dentistry, The University of Tennessee Health Sciences Center, Memphis, TN 38163, USA; 4Division of Head and Neck Surgical Oncology, The University of Tennessee Health Sciences Center, Memphis, TN 38163, USA; 5Department of Radiation Oncology, College of Medicine, The University of Tennessee Health Sciences Center, Memphis, TN 38163, USA

**Keywords:** dental oncology, oral cancer, oropharyngeal cancer, HPV, osteoradionecrosis, IMRT, xerostomia, trismus, mucositis, proton beam radiotherapy, medication-related osteonecrosis of the jaw, bone marrow transplant, haemopoietic stem cell transplant, immune checkpoint inhibitors, immunotherapy

## Abstract

This narrative review addresses the role of a dentist in the management of oncology patients, highlighting the oral complications that arise in head and neck radiation oncology patients and medical oncology patients. The prevention and management of these complications are discussed.

## 1. Introduction

This year, over 54,000 new cases of oral/pharyngeal cancer, as well as nearly 2 million new cancer cases, are projected to be diagnosed in the United States [1]. As dental practitioners, we are likely to encounter these oncology patients as pre-diagnosed, newly diagnosed, or post-oncologic treated patients. This narrative review will address the role of a dentist in the management of head and neck radiation oncology patients and medical oncology patients.


**Part 1: The Head and Neck Radiation Oncology Patient**


The majority of head and neck cancer cases include squamous cell carcinoma of the oral cavity and oropharynx (Figure 1 and Figure 2). These tumors are unified by a common anatomy and can be difficult to manage due to their proximity to vital structures. Head and neck cancer patients regularly undergo radiation therapy (RT) either as a primary or adjuvant therapy in combination with one or more modalities, such as chemotherapy and/or surgery. Treatment field planning is a critical part of RT management of head and neck cancer. The plans are drawn using computed tomography images to encompass the gross extent of the tumor, areas of potential microscopic involvement, high risk lymph nodes, patient motion, and setup error [2] (Figure 3 and Figure 4). Therapeutic RT standard fractionation for head and neck cancer entails two Gy per fraction, given five fractions per week, for 30–35 fractions [2].

There have been technological advances in radiation delivery techniques from conventional/2D techniques to 3D-conformal RT to intensity-modulated radiation therapy (IMRT)/proton beam radiation therapy (PBRT). These advanced techniques (IMRT and PBRT) enable improved tumor dose conformality and tissue/organ sparing capability [2,3,4,5]. Although IMRT and PBRT can reduce the risk of radiotherapy-related toxicities such as mucositis, trismus, xerostomia/hyposalivation, osteoradionecrosis (ORN) and abnormal development of orofacial structures, patients can still suffer from these complications, resulting in a decrease in quality of life [2,6,7,8,9,10,11].

### 1.1. Oral Mucositis

Mucositis is the inflammation of the mucosa, associated with thinning and apoptosis/necrosis of the mucosa (Figure 5). It is the most common acute complication of head and neck RT. The etiology of mucositis is the effect of radiation on mucosal cells. Radiation breaks the double-stranded DNA of the cells, resulting in lethal damage [12]. Oral mucositis (OM) is characterized by pain, erosions, ulcerations, odynophagia, secondary infections, reduced oral intake and, in severe cases, the necessity for a feeding tube or total parenteral nutrition. OM usually develops 2 weeks after the initiation of head and neck RT and may persist for 6 months after completion of RT. Almost all patients receiving head and neck radiation +/− chemotherapy develop oral mucositis.

Oral mucositis can be graded using the WHO toxicity scale:

Grade one—Soreness ± erythema.

Grade two—Erythema + Ulcers; patient can swallow food.

Grade three—Extensive erythema/ulceration; patient cannot swallow food.

Grade four—Extensive erythema/ulceration; oral nourishment impossible (life threatening).

Grades three and four are considered severe mucositis. Of the patients with severe oral mucositis, 70% require feeding tubes and 62% of these patients require hospitalization for total parenteral nutrition, intravenous analgesia and intravenous antibiotics [13]. Cessation of RT or a reduction in radiation dose is required by 35% of patients due to severe oral mucositis, which then increases the risk of tumor recurrence or treatment failure [13].

#### Prevention and Management of Oral Mucositis

Amifostine is a chemoradioprotective drug approved by the Food and Drug Administration (FDA) to help reduce the severity of mucositis in patients undergoing adjuvant chemo-RT for head and neck cancer. However, amifostine is limited in its use due to its associated severe side effects such as rapid hypotension, nausea, vomiting and a narrow therapeutic window related to its rapid systemic clearance [14,15]. The use of PBRT for head and neck cancer has been shown to reduce the rates of acute complications such as mucositis, nausea, dysgeusia and fatigue compared with IMRT [2]. Systematic reviews and meta-analyses on photobiomodulation (low-level laser therapy) in preventing and treating severe oral mucositis concluded that it significantly reduces the rate of this acute complication in chemotherapy and radiotherapy patients [16,17]. Cryotherapy has been shown to be effective in preventing severe oral mucositis, and it has been suggested that the combination of photobiomodulation and cryotherapy would be more beneficial for patients [18,19]. Topical analgesic rinses and anti-inflammatory and coating agents for pain control enable patients to eat, and antimicrobials aid patients at risk for infection. The Adapted Multinational Association of Supportive Care in Cancer (MASCC) and International Society of Oral Oncology (ISOO) clinical practice guidelines for oral mucositis are presented in Table 1 [20].

### 1.2. Trismus

Trismus is a well-known complication of radiation therapy [21,22] (Figure 6). It is defined as an inability to open the mouth or difficulty with mouth opening secondary to spasm of the muscles of mastication. Normal mouth opening (maximal interincisal opening) in the healthy population ranges from 36 to 55 mm. Measurements < 35 mm are considered to be trismus [23]. Trismus is associated with difficulty in many aspects of daily living such as eating, maintaining oral hygiene, receiving dental treatment, as well as impaired speech [24,25,26].

The prevalence of trismus among head and neck cancer patients managed with IMRT ranges from 4% to 77.3% [27,28,29,30,31]. Risk factors for the development of trismus include smaller pre-RT maximal interincisal opening and increased radiation dose to the muscles of mastication [7,32,33,34]. A study showed that patients with pre-RT MIO measurements of ≤40 mm were at risk of developing trismus, and the study by Teugh et al. reported that for every additional 10 Gy of radiation to the medial pterygoid muscle there was a 24% probability of developing trismus [7,34]. It is important to note that most patients typically present with limited mouth opening secondary to mucositis and not due to the radiation-related soft tissue fibrosis in trismus. Therefore, trismus can be either an acute or a chronic problem, depending on the etiology.

#### Prevention and Management of Trismus

Management of trismus is challenging, so practitioners should focus on prevention. The patient should be informed about this possible complication during a pre-RT dental appointment, as well as the need to perform regular mouth opening jaw exercises throughout the course of their RT [35,36]. Conservative management of trismus includes the use of jaw opening devices such as TheraBite, Dynasplint, corkscrews and stacked tongue depressors [37,38].

### 1.3. Xerostomia/Hyposalivation

Xerostomia is the most common long-term complication seen in head and neck cancer patients following RT [39,40]. The terms xerostomia and hyposalivation are frequently used interchangeably. However, xerostomia is a subjective assessment reported by patients as dry mouth and/or an altered taste. Hyposalivation is an objective assessment of reduction in salivary flow. This condition often limits the quality of life of patients regarding difficulty with mastication, speech, swallowing, halitosis, oral hygiene and dental caries.

The etiology of RT-induced xerostomia has been attributed to apoptosis or membrane damage-induced dysfunction of the salivary gland acinar cells and the inability of the damaged stem/progenitor cells to replace the apoptotic acinar cells [41,42]. The main risk factor for the development of RT-induced xerostomia is the dose of radiation to the salivary glands. Studies have reported a minimal salivary gland dysfunction at a mean dose (Dmean) of <10 Gy to the salivary gland, and at Dmean > 40 Gy there is complete salivary gland dysfunction [43,44,45]. Higher RT doses to the parotid glands predispose patients to xerostomia, although some patients may gradually recover within 24 months [8,43,45,46,47]. A study by Owosho et al. demonstrated that the rate of RT-induced xerostomia was highest within the first 6 months after treatment and is reduced at longer follow-up times [8]. The Quantitative Analysis of Normal Tissue Effects in the Clinic (QUANTEC) guidelines were developed to minimize the risk of salivary gland dysfunction either by: (i) one of the parotid glands receives a Dmean <~20 Gy; or (ii) both parotid glands receive a Dmean <~25 Gy. This has been shown to be effective in preventing the development of severe xerostomia [8,46,47].

#### Prevention and Management of RT-Induced Xerostomia

As noted earlier, amifostine is a chemoradioprotective drug approved by the FDA to help prevent moderate to severe xerostomia in patients undergoing adjuvant chemo-RT for head and neck cancer. However, amifostine is limited in its use because of its associated severe side effects listed above [14,15].

Xerostomia can be managed with parasympathomimetics such as pilocarpine and cevimeline. A systematic review and meta-analysis on the efficacy and safety of pilocarpine determined that pilocarpine was better than a placebo for the symptomatic treatment of radiation-induced xerostomia in patients with head and neck cancer [48]. Salivary flow stimulants and saliva substitutes or oral lubricants such as sugarless chewing gum, Biotène mouth oral rinse or moisturizing spray, ACT total care dry mouth mouthwash, XyliDent, sipping water and sucking on ice chips are also adjunct treatments. Patients with xerostomia should be prescribed an aggressive fluoride regimen and followed closely to help prevent and/or identify incipient dental caries. Adapted clinical recommendation guidelines by ISOO/MASCC/American Society of Clinical Oncology (ASCO) on xerostomia induced by nonsurgical cancer therapies are presented in Table 2 [49].

### 1.4. Osteoradionecrosis

ORN is the most severe complication encountered by head and neck cancer patients following radiation. ORN is defined as an area of exposed necrotic bone in a previously irradiated area that fails to heal over a period of 3–6 months [50,51] (Figure 7 and Figure 8). Cases of radiographic ORN with intact mucosa have been described [52,53]. The etiology of ORN has been attributed to the avascular effect of radiation to the bone, resulting in hypoxia, hypovascularity and hypocellularity [50]. The risk factors for the development of ORN can be broadly divided into local factors such as radiation dose, radiation field, proximity of tumor to bone, poor oral hygiene (periodontitis) and trauma (such as dental extractions and dentoalveolar surgery). Systemic factors include immunodeficient status, co-morbidities, smoking and alcohol use [10,54,55,56,57,58]. Radiation to the head and neck has been shown to increase periodontal attachment loss, resulting in mobile teeth, spontaneous exfoliation of teeth and non-exposed breakdown of alveolar bone.

ORN of the jaw is graded as follows:

Grade zero—Radiographic ORN with intact mucosa.

Grade one—Exposed necrotic bone without signs of infection for at least 3 months.

Grade two—Exposed necrotic bone with signs of infection or sequestrum, but not grades three and four.

Grade three—ORN resulting in pathologic fracture or ORN treated with successful surgical resection.

Grade four—ORN refractory to surgical resection.

The prevalence of ORN in the era of IMRT ranges from 4.3% to 6.8% [10,59,60]. Fortunately, the prevalence of ORN has decreased over recent decades. This reduction in prevalence is attributed to the advances in radiation delivery techniques. When comparing 3D conformal RT to IMRT, Tsai et al. showed a reduction in ORN from 13% to 6% [59]. However, it is important to note that a recent first study to evaluate the prevalence of ORN in head and neck cancer patients treated with PBRT reported a prevalence of 10.6% (13/122) [9]. The primary tumor sites for the 13 patients that developed ORN ranged from the base of tongue, tonsil, buccal mucosa, floor of mouth, hard palate, retromolar trigone and maxillary alveolus. All 13 patients reported no precipitating trauma (dental extraction or dentoalveolar surgery). This is of interest because PBRT is reported to demonstrate a greater tissue sparing capability than IMRT, which is attributed to the inert property of proton particles, known as *Bragg peak*, that allows deposition of its energy over a discrete range, thereby eliminating an exit dose to surrounding adjacent structures [61]. An earlier study on PBRT for the management of head and neck cancer patients reported lower rates of acute complications such as mucositis, nausea, dysgeusia and fatigue compared with IMRT [2].

High radiation dose remains the major risk factor for ORN. Studies have shown that radiation doses of >60 Gy to the bone are closely related to the risk of developing ORN [10,62]. A study evaluating the prevalence of ORN in head and neck cancer patients treated with IMRT reported that 96% of the ORN-affected sites received over 60 Gy (Dmax: 44.3 Gy–80.9 Gy) [10]. Another study by Singh et al. evaluating the prevalence of ORN following PBRT for head and neck cancer patients reported that all ORN-affected sites received over 60 Gy (Dmax: 61.8 Gy–81.2 Gy) [9].

#### Prevention and Management of Osteoradionecrosis

Managing ORN can be very challenging; thus, focusing on prevention is paramount. All patients who are scheduled to undergo RT for head and neck cancer should have a comprehensive dental evaluation. During this pre-RT dental evaluation, patients should be counseled on the possible oral complications of RT as well as preventive techniques. A comprehensive oral examination should be performed including panoramic and bitewing radiographs (and periapical radiographs when indicated). Prophylaxis (dental cleaning), topical fluoride application and detailed oral hygiene instructions are indicated at this time. Dental caries and periodontal disease should be treated as soon as possible. Indicated dental extractions (for non-restorable and unsalvageable periodontally involved teeth) should be completed at least 2 weeks prior to the commencement of RT. In addition, a prescription for high potency fluoridated toothpaste should be issued. Patients should be re-evaluated at mid-RT, at the completion of RT, periodically every 3 months for the first year and every 6 months, thereafter.

Invasive dental procedures such as extractions, deep root planing and implant placement should be avoided as much as possible after head and neck RT. If invasive dental procedures are necessary, the amount of radiation to the region of the jaw involved should be dosimetrically calculated. Extraction in an area of the jaw with a maximum dose point of >50 Gy is discouraged. Endodontic treatments are recommended. Decoronation of the tooth and root canal treatment of the retained root is encouraged.

There is no standard protocol in the management of ORN. Many different therapeutic approaches have been utilized based on the grade at presentation: chlorhexidine (wound care) rinse for local debridement; antibiotic and pain medications when indicated until the exposed necrotic bone is loose and can be passively removed; hyperbaric oxygen (which may require dives before and/or after the invasive dental procedure); pentoxifylline and tocopherol +/− antibiotics; photobiomodulation (low-level laser therapy); teriparatide; and surgical options (alveolectomy, maxillectomy or resection) [63,64,65,66]. In our experience, head and neck surgical oncologists now consider the use of pentoxifylline and tocopherol protocol as the typical first line management for ORN, and it has decreased the rate of major surgical resection and reconstruction.

### 1.5. Abnormal Development of Orofacial Structures

Radiotherapy to the head and neck of children can potentially lead to abnormal development of orofacial structures. Rhabdomyosarcoma (RMS) is the most common pediatric soft tissue sarcoma, accounting for 5–8% of all childhood malignancies, and the head and neck is the most common location for RMS [67]. Management of RMS includes surgery, chemotherapy, and radiotherapy.

The following late sequelae of RT-related toxicities to the orofacial structures in children have been reported: facial asymmetry secondary to jaw hypoplasia; tooth agenesis/hypodontia/oligodontia; and root agenesis/shortening/malformation [68,69,70]. Reports of these complications have persisted even in the era of IMRT [11]. The severity of the orofacial developmental abnormalities is related to the age of the child at treatment, the tumor/therapy site and the dose of radiation [11,71]. Younger children face greater risk and severity of orofacial developmental abnormalities. Kaste et al. reported that a radiation dose > 20 Gy to the orofacial region of children places them at a 4–10-fold higher risk of developing dental abnormalities [71]. Ameloblasts (tooth-forming progenitor cells) are damaged at a radiation dose of 10 Gy, and at 30 Gy tooth development is completely halted [72].

#### Prevention and Management of Abnormal Development of Orofacial Structures

Prevention of this late sequelae of RT for pediatric head and neck cancer may not be possible, as sparing these structures from the risk of radiation without compromising locoregional cancer control is challenging. However, a pediatric dentist should be integrated in the care of children with head and neck cancers for close surveillance and management of these complications.


**Part 2: The Medical Oncology Patient**


This second part of the review article will address the role of a dentist in the management of medical oncology patients.

## 2. Oncology Patients on Antiresorptive/Antiangiogenic Medications

The most common cancers in males and females in the United States are prostate and breast cancers, respectively [1]. Advanced stages of many malignancies are associated with metastases to bone. Multiple myeloma and metastatic bone disease may necessitate the use of bone-modifying agents, such as antiresorptive medications, including intravenous bisphosphonates (pamidronate and zoledronate) and humanized monoclonal anti-RANK ligand (denosumab). These bone-modifying agents help in the prevention of skeletal-related events such as: spinal bone compression, bone fracture and surgery, as well as radiotherapy to bone [73,74]. These antiresorptive medications alongside other oncologic medications, including tyrosine kinase inhibitors (sunitinib and imatinib), monoclonal antibodies (bevacizumab), mTOR inhibitors (everolimus), anti-CTLA4 (ipilimumab), fusion protein (ziv-aflibercept) and immunosuppressants (methotrexate and corticosteroids), are reported to be associated with medication-related osteonecrosis of the jaw (MRONJ) [75,76,77,78,79,80,81,82].

MRONJ is defined as an area of exposed bone or bone that can be probed through an intraoral or extraoral fistula(e) in the maxillofacial region that has persisted for more than 8 weeks in an area not previously irradiated or metastases to the bone, in a patient with a history of treatment with antiresorptive therapy alone or in combination with immune modulators or antiangiogenic medication [82]. The etiology of MRONJ is multifactorial, including bone remodeling inhibition, inflammation or infection, angiogenesis inhibition, innate or acquired immune dysfunction and genetic predisposition [82].

The risk factors for the development of MRONJ can be broadly divided into three local factors: dentoalveolar trauma (such as extractions and dentoalveolar surgery), poor oral hygiene (periodontitis) and anatomic location (mandible > maxilla) [83,84,85]. Systemic factors include age (older patients), gender (F > M), co-morbidities, smoking and corticosteroids [85,86]. The risk of developing MRONJ after a tooth extraction among cancer patients taking antiresorptive medications ranges from 1.6% to 14.8% [87,88,89]. It is important to note that cases of MRONJ without a precipitating dental trauma have been reported as occurring spontaneously [83].

MRONJ is staged as follows [82]:

Stage zero—Clinical/radiographic MRONJ with intact mucosa, no exposed necrotic bone.

Stage one—Exposed necrotic bone or fistula that probes to bone without signs of infection.

Stage two—Exposed necrotic bone or fistula that probes to bone with signs of infection or sequestrum.

Stage three—Exposed necrotic bone or fistula that probes to bone with signs of infection or sequestrum. In addition, one or more of the following: exposed necrotic bone extending beyond the region of alveolar bone, pathologic fracture, extraoral fistula, oral antral/oral–nasal communication and osteolysis extending to the inferior border of the mandible or sinus floor.

The risk of MRONJ is significantly greater in cancer patients receiving antiresorptive therapy compared with patients receiving antiresorptive therapy for osteoporosis. This disparity is due to differences in the dose, frequency of use and mode of administration of the antiresorptive medications. The risk of MRONJ amongst oncology patients taking zoledronate ranges from 0 to 18%, with most studies reporting < 5%, and that for oncology patients taking denosumab ranges from 0 to 6.9%, with most studies reporting < 5% [82,90,91,92]. Oncology patients on denosumab are at risk of developing MRONJ earlier. A study reported that cancer patients using denosumab developed MRONJ earlier or at fewer doses compared with cancer patients using intravenous bisphosphonates (pamidronate or zoledronate) [83].

### 2.1. Prevention and Management of MRONJ

The role of pre-medication dental evaluation (PMDE) in prevention of MRONJ has been shown to be very effective in reducing the incidence of MRONJ among oncology patients using antiresorptive/antiangiogenic medications. The study by Owosho et al. showed a 12-fold decrease in the incidence of MRONJ among oncology patients who received PMDE compared with patients who did not receive PMDE [83]. Studies by Bramati et al. and Bonacina et al. with 212 and 282 patients, respectively, demonstrated a reduction in the incidence of MRONJ from 8.6% and 10.8%, respectively, to 0% after implementing PMDE [93,94]. Similar studies by Dimopoulos et al. and Ripamonti et al. reported similar outcomes of lesser incidence of MRONJ in patients that received PMDE [95,96].

PMDE entails patient education, comprehensive dental evaluation, and completion of recommended dental treatment [83]. During the PMDE visit, oncology patients should be informed about the potential oral complication of MRONJ and the need to avoid any form of invasive dental procedures after commencing the antiresorptive/antiangiogenic medications. In addition, patients are provided oral hygiene instructions and nutritional counselling. A comprehensive dental examination should be performed including panoramic and bitewing radiographs, (and periapical radiographs when indicated). Prophylaxis (dental cleaning) and topical fluoride application are indicated as well. Dental caries and periodontal disease should be treated as soon as possible. Indicated dental extractions (non-restorable and periodontally involved teeth) should be completed at least 2–3 weeks prior to the commencement of antiresorptive/antiangiogenic medications. In addition, a prescription for high potency fluoridated toothpaste should be provided. Patients should be periodically evaluated every 3 months for the first year and every 6 months, thereafter. The American Association of Oral and Maxillofacial Surgeons’ position paper on MRONJ—2022 Update states that the clinical utility of drug holidays to mitigate MRONJ risk in patients undergoing dentoalveolar surgery remains controversial [82].

There are different management approaches to MRONJ based on the stage at presentation: chlorhexidine (wound care) rinse for local debridement and antibiotic and pain medications when indicated until the exposed necrotic bone is loose and can be passively removed. The use of other non-surgical options, such as pentoxifylline and tocopherol +/− antibiotics; photobiomodulation (low-level laser therapy); and teriparatide have shown great promise as effective management adjuncts [97,98,99,100,101]. Due to the unpredictable, protracted course and varying success rates of non-surgical approaches in the management of MRONJ, surgical options (alveolectomy, maxillectomy or resection [marginal or segmental]) should be considered in the management of MRONJ. Surgical options have reported high success rates [82]. Adapted MASCC/ISOO/ASCO clinical practice guidelines for MRONJ are presented in Table 3 [102].

### 2.2. Hematopoietic Stem Cell Transplant (HSCT)

Hematopoietic stem cell transplant (HSCT) is widely used in the management of malignant and non-malignant hematologic diseases, such as leukemia, lymphoma, multiple myeloma, aplastic anemia and immune deficiency disorders for the purpose of immune system reconstitution following conditioning. This form of oncologic therapy renders patients severely immunocompromised and at risk of bleeding prior to, during and long after the therapy, secondary to cytopenia and thrombocytopenia [103]. During this period of immunosuppression, patients are at an increased risk of infection with potential for systemic spread. Poor oral health predisposes patients to bacteremia, and odontogenic infections can become severe enough to contribute to morbidity and mortality [104]. HSCT (conditioning) also causes mucositis and salivary gland dysfunction (xerostomia) [105,106]. Oral mucositis usually begins a week after initiation of conditioning and continues for two weeks after the end of conditioning. Another complication that arises after allogeneic HSCT is chronic graft-versus-host disease (cGVHD), which occurs within several months after HSCT in a high percentage of patients. cGVHD affects multiple organs, including the skin, eyes, gastrointestinal tract-oral cavity, lungs and liver [107]. Oral cGVHD is characterized by ulcers, erythematous and lichenoid lesions [108].

For the aforementioned reasons, pre-HSCT dental clearance with an established dental clearance protocol should be incorporated in the management of these patients. The National Cancer Institute, Centers for Disease Control and Prevention, the joint taskforce of the Multinational Association for Supportive Care in Cancer with the International Society of Oral Oncology and the European Society for Blood and Marrow Transplantation have all recommended dental clearance prior to undergoing HSCT. Pre-HSCT includes a comprehensive oral evaluation and completion of recommended dental treatment. A comprehensive oral examination should be performed, including panoramic and bitewing radiographs (and periapical radiographs when indicated), for evaluation of dental caries, periapical/periodontal/bone pathologies, and the status of the third molars. Following the oral evaluation, the recommended dental therapies such as caries control, endodontic therapy, periodontal therapy (prophylaxis, scaling and root planing) and topical fluoride application are to be completed as soon as possible. Indicated dental extractions for non-restorable and unsalvageable periodontally involved teeth should be completed at least 2 weeks prior to the commencement of HSCT/high-dose chemotherapy. In addition, a prescription for high potency fluoridated toothpaste should be provided.

The timing of dental procedures during the pre-HSCT dental clearance in these patients should be coordinated with the medical team. The patient’s hematologic laboratory result should be reviewed for complete blood count (white blood cell count, absolute neutrophil count [ANC] and platelet count) before performing any dental procedures to avert the risk of infection and bleeding. A patient with severe neutropenia (ANC: 0–499) is at risk of life-threatening infection and should be isolated and hospitalized. The only treatment that should be performed is palliative care under antibiotic prophylaxis, oral antimicrobial rinse, and post-procedural antibiotic regimen for a week. No routine care should be completed on these patients. Patients with moderate neutropenia (ANC: 500–999) are at a moderate risk for infection. They should be given antibiotic prophylaxis, oral antimicrobial rinse before any dental procedure and post-procedural antibiotic regimen for 5 days. While patients with mild neutropenia (ANC: 1000–1500) are at a mild risk for infection, they can receive routine dental care without antibiotic prophylaxis (except in instances of major surgery, such as multiple dental extractions) and post-procedural antibiotic regimen for 5 days. Patients with a platelet count of <50,000 should not receive any invasive dental procedure due to the risk of uncontrollable bleeding. At platelet counts < 20,000, the patient may bleed spontaneously.

The acute complications of oral mucositis and xerostomia can be managed as described earlier in the first part of this review article.

### 2.3. Immune Checkpoint Inhibitors (Immunotherapy)

Immune checkpoint inhibitors such as anti-CTLA4 (Ipilimumab) and PD-1 inhibitors (Pembrolizumab and Nivolumab) are now used in the management of several advanced cancers. PD-1 inhibitors have been approved by the FDA for a subset of patients with advanced head and neck cancer. The following oral toxicities of these immune checkpoint inhibitors have been reported: xerostomia, dysgeusia, mucositis, lichenoid reaction and MRONJ [78,109,110,111].

## 3. Conclusions

As dental practitioners, we play a crucial role in the management of the head and neck radiation oncology/medical oncology patients, from making the initial diagnosis of the oral/oropharyngeal cancer to the identification and management of the associated oncologic therapy complications. Pre-radiation therapy/antiresorptive/antiangiogenic medications and pre-HSCT dental evaluation is an essential management step in the prevention and management of these complications. Each comprehensive head and neck/medical oncology team should have a dentist integrated in the care of these oncology patients.

## Figures and Tables

**Figure 1 dentistry-11-00136-f001:**
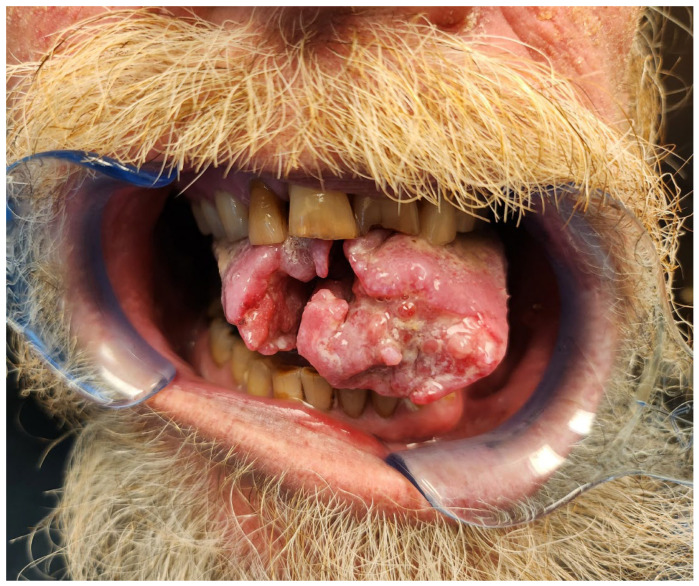
Squamous cell carcinoma of the tongue in a 59-year-old patient with a significant history of tobacco smoking.

**Figure 2 dentistry-11-00136-f002:**
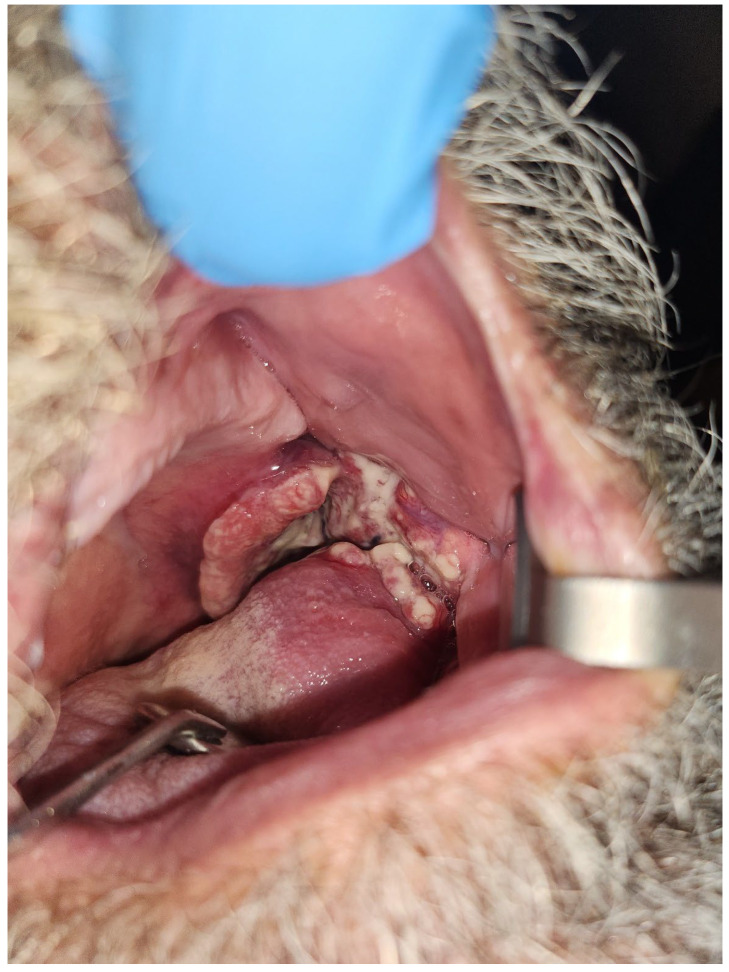
Human papillomavirus negative squamous cell carcinoma of the left tonsil in a 61-year-old patient with a significant history of tobacco smoking.

**Figure 3 dentistry-11-00136-f003:**
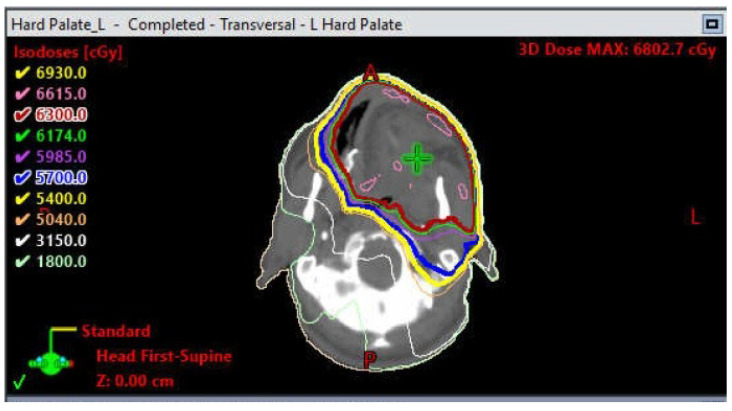
Treatment field plan for an oncology patient who received adjuvant radiation therapy for a high-grade adenocarcinoma of the palate.

**Figure 4 dentistry-11-00136-f004:**
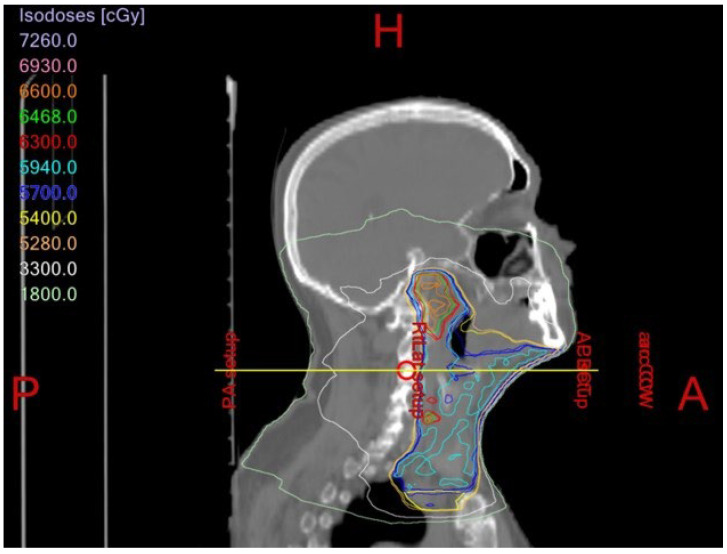
Treatment field plan for an oncology patient who received adjuvant radiation therapy for squamous cell carcinoma of the face.

**Figure 5 dentistry-11-00136-f005:**
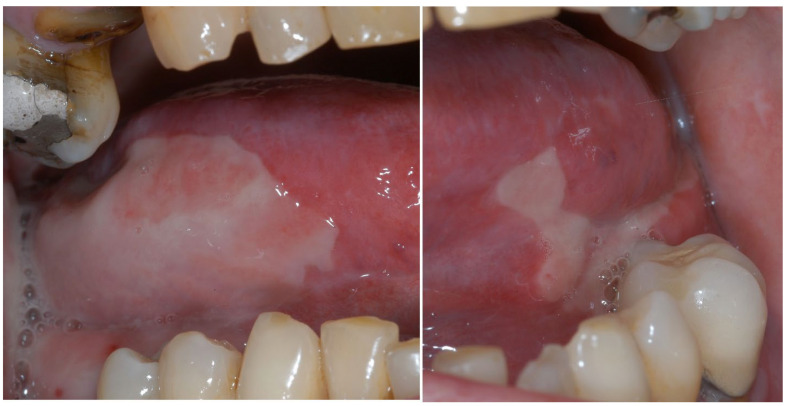
Oral mucositis in an oncology patient receiving adjuvant radiation therapy for oropharyngeal cancer.

**Figure 6 dentistry-11-00136-f006:**
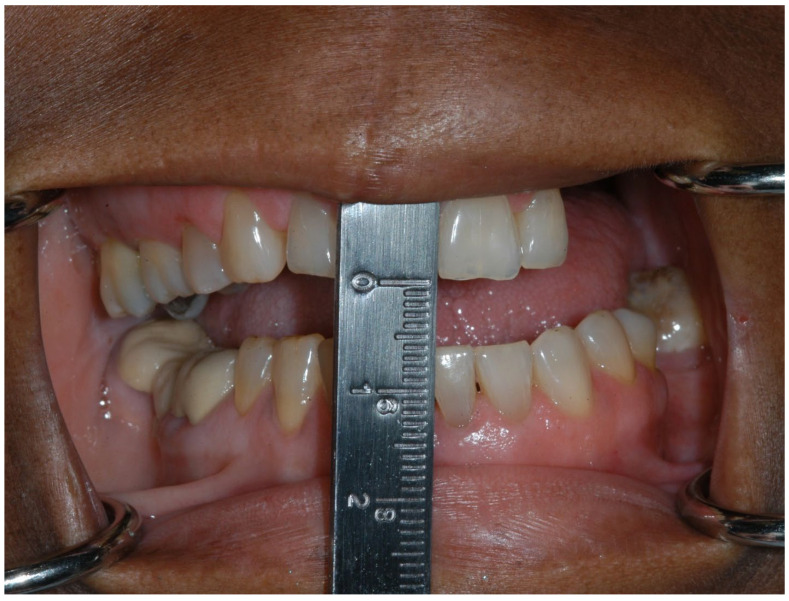
Trismus (maximal interincisal opening of 6 mm) in an oncology patient who received adjuvant radiation therapy.

**Figure 7 dentistry-11-00136-f007:**
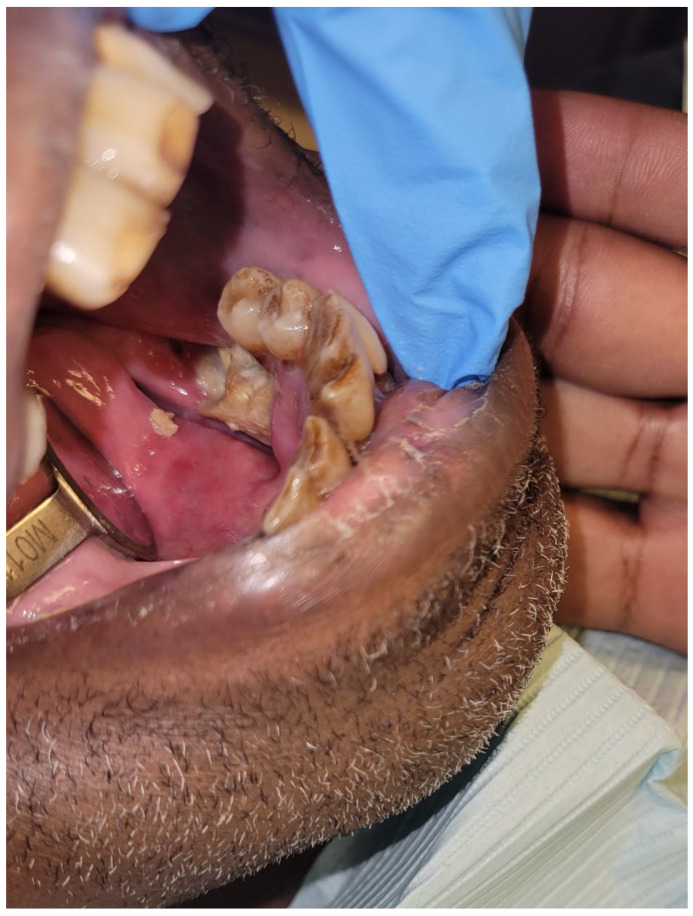
Osteoradionecrosis in an oncology patient who received adjuvant radiation therapy for left tonsil squamous cell carcinoma. Clinical image shows an exposed necrotic bone in the region of missing tooth #19. Patient reported that tooth #19 spontaneously exfoliated.

**Figure 8 dentistry-11-00136-f008:**
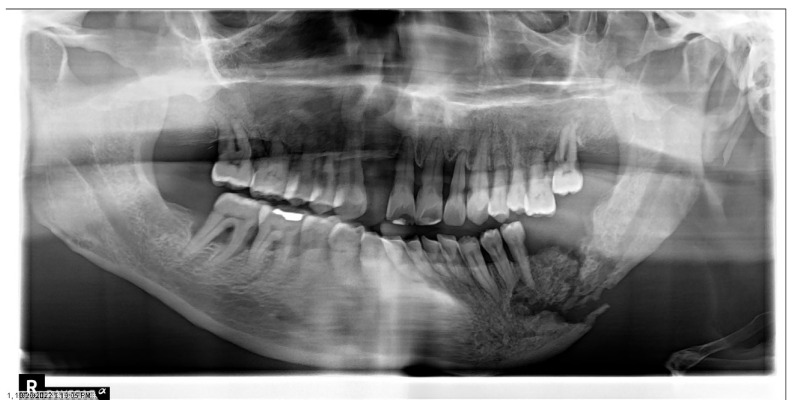
Osteoradionecrosis in an oncology patient who received adjuvant radiation therapy for left tonsil squamous cell carcinoma. Panoramic radiograph reveals pathologic fracture of the mandible. Radiograph of the same patient in Figure 7.

**Table 1 dentistry-11-00136-t001:** Adapted MASCC/ISOO clinical practice guidelines for oral mucositis [20].

Section	Guideline Statement
Basic oral care	The panel suggests that implementation of multiagent combination oral care protocols (MCOCP) is beneficial for the prevention of oral mucositis during chemotherapy, head and neck radiotherapy and hematopoietic stem cell transplant (HSCT). MCOCP—these protocols serve to increase the awareness of patients and staff of the importance of good oral hygiene that may lead to fewer and less severe oral complications; typically, the protocols involve recommendations about the timing, frequency and products used, which include various combinations of bland mouth rinses, toothbrushes and flossing procedures.
Anti-inflammatory agents	The panel recommends benzydamine mouthwash for the prevention of oral mucositis in patients with H and N cancer receiving a moderate-dose RT (<50 Gy).
	The panel suggests the use of benzydamine mouthwash for the prevention of oral mucositis in patients with head and neck cancer who receive chemoradiation therapy.
Photobiomodulation therapy	The panel recommends the use of intraoral photobiomodulation therapy using low-level laser therapy for the prevention of oral mucositis in adult patients receiving radiotherapy ± chemotherapy for head and neck cancers or HSCT conditioned with high-dose chemotherapy ± total body irradiation. Safety considerations unique to patients with oral cancer should be considered.
Cryotherapy	The panel recommends using oral cryotherapy to prevent oral mucositis in patients undergoing autologous HSCT when the conditioning includes high dose melphalan.
	The panel recommends using 30 min of oral cryotherapy to prevent oral mucositis in patients receiving bolus 5-FU chemotherapy during the infusion.
Antimicrobials, coating agents, anesthetics and analgesics	Topical morphine 0.2% mouthwash is suggested for the treatment of oral mucositis-associated pain in patients with head and neck cancer who receive chemoradiation therapy.
Growth factors and cytokines	The use of KGF-1 intravenously is recommended for the prevention of oral mucositis in patients with hematologic cancer undergoing autologous HSCT with a conditioning regimen that includes high-dose chemotherapy and total body irradiation.
Natural and miscellaneous	Honey is suggested for the prevention of oral mucositis in patients with head and neck cancer who receive chemoradiation therapy.

**Table 2 dentistry-11-00136-t002:** Adapted clinical recommendation guidelines by ISOO/MASCC/ASCO on xerostomia induced by nonsurgical cancer therapies [49].

**Clinical question 1**. What is the efficacy of available pharmacologic and nonpharmacologic interventions (including the effects of radiation dose, type and regimen) for the prevention of xerostomia induced by nonsurgical cancer therapies?
**Recommendation 1.1.** IMRT should be used to spare major and minor salivary glands from a higher dose of radiation to reduce the risk of salivary gland hypofunction and xerostomia in patients with head and neck cancer *(type: evidence-based; evidence quality: high; strength of recommendation: strong).*
**Recommendation 1.2.** Other radiation modalities that limit cumulative dose to and irradiated volume of major and minor salivary glands as or more effectively than intensity-modulated radiation therapy may be offered to reduce salivary gland hypofunction and xerostomia *(type: informal consensus; evidence quality: low; strength of recommendation: strong).*
**Recommendation 1.3.** Acupuncture may be offered during radiation therapy for head and neck cancer to reduce the risk of developing xerostomia *(type: evidence-based; evidence quality: intermediate; strength of recommendation: moderate).*
**Recommendation 1.4.** Systemic administration of the sialogogue bethanechol may be offered during radiation therapy for head and neck cancer to reduce the risk of salivary gland hypofunction and xerostomia *(type: evidence-based; evidence quality: low; strength of recommendation: weak).*
**Clinical question 2.** What is the efficacy of available pharmacologic and nonpharmacologic interventions for the management of xerostomia induced by nonsurgical cancer therapies?
**Recommendation 2.1.** Topical mucosal lubricants or saliva substitutes (agents directed at ameliorating xerostomia and other salivary gland hypofunction-related symptoms) may be offered to improve xerostomia induced by nonsurgical cancer therapies *(type: evidence-based; evidence quality: intermediate; strength of recommendation: strong).*
**Recommendation 2.2.** Gustatory and masticatory salivary reflex stimulation by sugar-free lozenges, acidic (nonerosive and sugar-free special preparation if dentate patients) candies or sugar-free, nonacidic chewing gum may be offered to produce transitory increased saliva flow rate and transitory relief from xerostomia by stimulating residual capacity of salivary gland tissue *(type: evidence-based; evidence quality: intermediate; strength of recommendation: moderate).*
**Recommendation 2.3.** Oral pilocarpine, and cevimeline where available, may be offered after radiation therapy in patients with head and neck cancer for transitory improvement of xerostomia and salivary gland hypofunction by stimulating residual capacity of salivary gland tissue. However, improvement of salivary gland hypofunction may be limited *(type: evidence-based; evidence quality: high; strength of recommendation: strong).*
**Recommendation 2.4.** Acupuncture may be offered after radiation therapy in patients with head and neck cancer for improvement of xerostomia *(type: evidence-based; evidence quality: low; strength of recommendation: weak).*
**Recommendation 2.5.** Transcutaneous electrostimulation or acupuncture-like transcutaneous electrostimulation of the salivary glands may be offered after radiation therapy in patients with head and neck cancer for improvement of salivary gland hypofunction and xerostomia *(type: evidence-based; evidence quality: low; strength of recommendation: weak)*

**Table 3 dentistry-11-00136-t003:** Adapted MASCC/ISOO/ASCO clinical practice guidelines for MRONJ [102].

**Clinical Question 2.** What steps should be taken to reduce the risk of MRONJ?
**Recommendation 2.1**: Coordination of care: for patients with cancer who are scheduled to receive a BMA in a nonurgent setting, oral care assessment (including a comprehensive dental, periodontal and oral radiographic exam when feasible to do so) should be undertaken before initiating therapy. Based on the assessment, a dental care plan should be developed and implemented. The care plan should be coordinated between the dentist and the oncologist to ensure that medically necessary dental procedures are undertaken before the initiation of the BMA. Follow-up by the dentist should then be performed on a routine schedule, such as every 6 months once therapy with a BMA has commenced *(Type: evidence based; Evidence quality: low/intermediate; Strength of recommendation: moderate).*
**Recommendation 2.2**. Modifiable risk factors: members of the multidisciplinary team should address modifiable risk factors for MRONJ with the patient as early as possible. These risk factors include poor oral health, invasive dental procedures, ill-fitting dentures, uncontrolled diabetes mellitus and tobacco use *(Type: formal consensus; Evidence quality: insufficient; Strength of recommendation: moderate).*
**Recommendation 2.3**. Elective dentoalveolar surgery: elective dentoalveolar surgical procedures (e.g., nonmedically necessary extractions, alveoloplasties and implants) should not be performed during active therapy with a BMA at an oncologic dose. Exceptions may be considered when a dental specialist with expertise in the prevention and treatment of MRONJ has reviewed the benefits and risks of the proposed invasive procedure with the patient and the oncology team *(Type: evidence based; Evidence quality: intermediate; Strength of recommendation: moderate).*
**Recommendation 2.4**. Dentoalveolar surgery follow-up: if dentoalveolar surgery is performed, patients should be evaluated by the dental specialist on a systematic and frequently scheduled basis (e.g., every 6 to 8 weeks) until full mucosal coverage of the surgical site has occurred. Communication with the oncologist regarding the status of healing is encouraged, particularly when considering future use of BMA *(Type: formal consensus; Evidence quality: insufficient; Strength of recommendation: moderate).*
**Recommendation 2.5**. Temporary discontinuation of BMAs before dentoalveolar surgery: for patients with cancer who are receiving a BMA at an oncologic dose, there is insufficient evidence to support or refute the need for discontinuation of the BMA before dentoalveolar surgery. Administration of the BMA may be deferred at the discretion of the treating physician, in conjunction with discussion with the patient and the oral health provider *(Type: informal consensus; Evidence quality: insufficient; Strength of recommendation: weak).*
**Clinical Question 4**. How should MRONJ be managed?
**Recommendation 4.1**: Initial treatment of MRONJ: conservative measures comprise the initial approach to treatment of MRONJ. Conservative measures may include antimicrobial mouth rinses, antibiotics if clinically indicated, effective oral hygiene and conservative surgical interventions, such as removal of a superficial bone spicule *(Type: formal consensus; Evidence quality: insufficient; Strength of recommendation: moderate).*
**Recommendation 4.2**: Treatment of refractory MRONJ: aggressive surgical interventions (e.g., mucosal flap elevation, block resection of necrotic bone or soft tissue closure) may be used if MRONJ results in persistent symptoms or affects function despite initial conservative treatment. Aggressive surgical intervention is not recommended for asymptomatic bone exposure. In advance of the aggressive surgical intervention, the multidisciplinary care team and patient should thoroughly discuss the risks and benefits of the proposed intervention *(Type: formal consensus; Evidence quality: insufficient; Strength of recommendation: weak).*
**Clinical Question 5**. Should BMAs be temporarily discontinued after a diagnosis of MRONJ has been made?
**Recommendation 5**. For patients who are diagnosed with MRONJ while being treated with BMAs, there is insufficient evidence to support or refute the discontinuation of the BMAs. Administration of the BMA may be deferred at the discretion of the treating physician, in conjunction with discussion with the patient and the oral health provider *(Type: formal consensus; Evidence quality: insufficient; Strength of recommendation: weak).*

## Data Availability

No data.

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
