# Peer review of "The Role of Dental Practitioners in the Management of Oncology Patients: The Head and Neck Radiation Oncology Patient and the Medical Oncology Patient"

_dentistry, 2023, doi:10.3390/dj11050136_

Round 1

Reviewer 1 Report

The article discusses the role of dentists in the management of cancer, head and neck, and oncology patients in general.

The text is a simple review of the literature, and lacking is a lot of current information from worldwide guidelines in the management of the cancer patient group. The division by groups into more articles would be very opportune, since aggregating the care of irradiated patients in the head and neck region, with solid tumors, oncohematologic diseases, and hematopoietic stem cell transplantation into one article is not feasible.

If the purpose of the article is to provide guidelines to assist dentists in the management of patients with cancer, several clinical figures are dispensable, because the focus is on guidelines and not on diagnosis. 

There is a lack of information overall in the clinical management of cancer patients in the various oral complications mentioned in the article. There is also a lot of current information that was not covered by the authors.

There is an excess of self-citation.

Author Response

We would like to thank the reviewer for taking the time to review our manuscript. An adapted guideline from MASCC/ISOO on oral mucositis, xerostomia and MRONJ now included.

Thank you.

Dr. Adepitan Owosho

Reviewer 2 Report

This review requires much more effort in terms of scientific methodology. It does not has any materials and methods section, how was the literature reviewed, which articles were considered, which was the selected data...and many more. I know it is not a a systematic review, however there are minimum requirements for a narrative review.

Author Response

We would like to thank the reviewer for taking the time to review our manuscript. As rightly stated this is not a systematic review. It is a narrative review based on the authors experience to help practicing dentist in managing oncology patients. Clinical practice guidelines from MASCC/ISOO are now included.

Thank you.

Dr. Adepitan Owosho

Reviewer 3 Report

Very interesting work comprehensive and significant in scope. describes the importance of knowledge of oral pathology. I would like to contribute to your work by inserting in your article the following reference which is related to your work. Explaining how even the laser can be useful in the management of these lesions.

Laser photobiomodulation (Pbm)—a possible new frontier for the treatment of oral cancer: A review of in vitro and in vivo studies

Del Vecchio, A.Tenore, G.Luzi, M.C., ...Pergolini, D.Romeo, U.

Author Response

We would like to thank the reviewer for the kind words. The article mentioned is now referenced.

Thank you.

Dr. Adepitan Owosho

Reviewer 4 Report

This paper is of great value to dental practitioners treating oncology patients and covers the full spectrum of potential problems in the management of this group of patients.

However, to make it more "up-to-date" I would remove references older than 10 years (56/113) as progress in oncological treatment, especially radiotherapy, influences types of complications and methods of treatment.  

Author Response

We would like to thank the reviewer for the kind words. We have reduced the number of references to more recent ones. However, some old pertinent references are still in place.

Thank you.

Dr. Adepitan Owosho

Round 2

Reviewer 1 Report

There have been several corrections, and improvement of the scientific quality. But, there are still many self-citations.

Author Response

We would like to thank the reviewer for reviewing our manuscript. We have deleted 3 self-citation references of the first author. We believe the remaining self-cited references are essentially referenced in the cited discussions.

Thank you for your time.

Dr. Adepitan A. Owosho

Reviewer 2 Report

My only concern keeps on the work not being related to a proper review structure....It looks like more like guidelines. If that so, I would recommend the authors to change the paper accordingly.